# Bedouin Adolescents during the Iron Swords War: What Strategies Help Them to Cope Successfully with the Stressful Situation?

**DOI:** 10.3390/bs14100900

**Published:** 2024-10-04

**Authors:** Haled Al-Said, Orna Braun-Lewensohn

**Affiliations:** 1Educational Counseling, Kaye Academic College of Education, P.O. Box 4301, Beer-Sheva 8414201, Israel; 2Conflict Management & Resolution Program, Ben-Gurion University of the Negev, P.O. Box 653, Beer-Sheva 8410501, Israel; ornabl@bgu.ac.il

**Keywords:** coping resources, Bedouin society, Iron Swords War

## Abstract

This study examined the emotional reactions and coping resources of Bedouin teenagers (ages 12–18) following the events of 7 October 2023 and the subsequent Iron Swords War between Israel and Hamas forces in Gaza. This study focused on the relationships between subjective exposure, personal sense of coherence, community resilience, and the use of different coping strategies, on the one hand, and anxiety, anger, and psychological distress, on the other. During 2024, 762 participants completed the study questionnaire, which addressed the different study variables. The findings revealed significant differences between the boys and the girls. Specifically, the girls reported higher levels of subjective exposure and emotional reactions. Personal sense of coherence was found to be a significant protective factor. In contrast, the use of nonproductive coping strategies increased distress. This study underscores the need for intervention programs tailored for this society and by gender, as well as training for educational professionals, to help them to identify and treat emotional reactions to stress in ways that take into consideration the Bedouin cultural context. This study enhances our understanding of how Bedouin teenagers cope with stressful situations and crises.

## 1. Introduction

### 1.1. Background

On 7 October 2023, the Hamas organization, which controls Gaza, launched a broad, surprise attack against the state of Israel, aimed particularly at the communities near the border with Gaza and at both military forces and innocent civilians. During this attack, 1200 Israeli citizens were murdered and 240 were kidnapped, with no distinctions made between individuals of different genders, ethnicities, or ages [1]. This event led Israel to fight the Iron Swords War, which continues as of the time of this writing [2]. At the start of the war, the Israeli government evacuated the residents of 30 communities in the south of the country and, after the northern front heated up, evacuated the residents of 21 additional communities. On 9 November 2023, a temporary ceasefire was declared, during which 105 hostages were released from Gaza [3]. Bedouin Arab society, which is the focus of this study, has lost 24 of its own, including 4 children, 2 teenagers, and 4 soldiers. Among the 128 hostages still held in Gaza are 4 Bedouin hostages, in addition to another Bedouin hostage who was mistakenly killed by the Israeli army. The hostages and the fallen are from well-known families in the Bedouin community and their stories resonate deeply and painfully within the community, particularly among teenagers. The fallen include two parents of large families (some of whose children are teenagers), the son of a family of teachers, teenagers who were active in their youth group, and a soldier-screenwriter who worked with teenagers [4].

By the middle of April 2024, 197 warnings of incoming missiles had been recorded in the Negev region [5]. The Bedouin population in the Negev lives in 7 recognized villages and 47 unrecognized villages (about 30% of the population live in unrecognized villages). The 47 unrecognized Bedouin villages are located in areas classified as “open areas”, which are not protected by Iron Dome anti-missile batteries. In the absence of protective spaces and permanent buildings, there have been high rates of direct hits from missile-fire, which have carried a price in terms of human life and property.

### 1.2. Teenagers in the Bedouin Arab Community in Southern Israel

The Bedouin Arab population numbers 279,300 people [6] and is a very young population; in 2020, over half of the population was aged 0–19. This community faces various economic, political, and social challenges [7], being the poorest group in Israel with significant gaps in all socioeconomic indicators. Bedouin villages, especially unrecognized ones, are characterized by poor physical infrastructure, very limited access to services, underdevelopment, and extreme poverty [8]. Four out of five Bedouin children live below the poverty line. Bedouin teenagers suffer from a severe lack of educational infrastructure, both formal and informal. The quality of teaching is low and there is a shortage of mental-health professionals. Consequently, dropout rates are high; 23% of students do not complete 12th grade. The integration rates of young Bedouins in higher education, vocational training, and the labor market is significantly low compared to Jewish society, and even lower than the rates among general Arab society. Violence and crime pose a significant problem, with 75% of respondents in a study conducted among Bedouin youth reporting personal insecurity due to violence in their locality. The reasons for this include harsh living conditions, uncertainty, power struggles over limited resources, and a lack of supportive frameworks. Conflicts between clans, deterioration in adult authority, and a lack of young leadership are significant factors contributing to violence and crime in Bedouin society. During times of political crisis, Bedouin teenagers have exhibited higher levels of anger and anxiety than other teenagers [9].

These events of political violence likely imposed an additional emotional burden on Bedouin Arab adolescents, due to the complex and confusing political situation they were forced to confront. This situation created a deep internal conflict within them. On one hand, many felt anger toward the Israeli leadership, which ordered the military to attack Gaza. For some of these adolescents, the conflict was particularly personal, as they had close relatives in the Gaza Strip (this information was received in personal communication from the Welfare Office of Rahat Municipality on 12 June 2014). This led to feelings of worry, anxiety, and frustration in the face of their inability to help their relatives [10].

On the other hand, the adolescents themselves were living under constant threat of missiles falling on their city, Rahat. This reality created a troubling paradox: while they emotionally identified with the Palestinians in Gaza, they themselves were victims of the same system of violence. This situation deepened their sense of confusion and internal conflict, as they had to deal with real physical danger alongside complex moral and emotional dilemmas.

Moreover, the unique position of the Bedouin-Arab community in Israel—caught between the hammer and the anvil of conflicting identities and loyalties—added another layer of complexity to the traumatic experience of these adolescents. All these factors together created a complicated and challenging emotional reality, which demanded significant mental strength for coping and adaptation, while also highlighting the multifaceted impact of the conflict on this particular community.

### 1.3. Strategies for Coping with Stressful Situations

We can define the term *coping* as the cognitive and behavioral-emotional efforts of an individual to deal with an external or internal demand that that individual perceives as weighty or difficult [11]. The literature [12] refers to two types of strategies for coping with stress: strategies that are focused on problem-solving and strategies that are focused on emotion. Strategies focused on problem-solving include active activities, such as planning, active suppression and restraint, the seeking of instrumental and emotional assistance, a positive outlook, acceptance of the situation, and choosing from among options. The goal of these strategies is to change or treat the issue that is causing the stress [13,14]. This type of strategy is considered effective when the source of the stress is perceived as something that the individual can control and the situation that is being experienced is perceived as changeable [15].

Coping strategies that are focused on emotion include emotional actions and reactions, such as physiological arousal, the release of emotions, denial or acceptance of the event, turning to religion, behavioral or cognitive disconnection, and the use of alcohol, or, in contrast, humor. This type of coping is aimed at regulating emotional reactions to the stressful situation. This type of coping strategy is considered effective for decreasing the sense of stress, particularly in the context of events that are perceived as being beyond the individual’s control [12]. However, in many contexts, emotional coping strategies are considered and referred to as nonproductive coping strategies [16].

A study conducted in 2020 found that Arab adolescents in situations of political pressure tended to use ineffective strategies, such as blaming, keeping the problems to themselves, and not sharing, as well as practices to release pressure, such as turning to religion or writing personal diaries. These adolescents expressed a concern that sharing with parents or teachers might harm them and their future [17].

### 1.4. Sociodemographic Factors

The research literature describes a variety of sociodemographic factors that make substantial contributions to the diminishment of stress reactions and to mental health. These include age [7,18] and gender [19,20].

#### 1.4.1. Age

More than a few studies have found a relationship between age and the intensity of an individual’s reactions to a stressful event [21]. Some of those studies reported a difference between younger teenagers and older teenagers, in terms of reactions to stress in the context of exposure to a terror attack. Specifically, the younger teenagers experienced more stress symptoms following the stressful event, including stomachaches and depression, as compared to older teenagers [22,23]. However, other research did not find any differences between the reactions of younger and older teenagers to a stressful terror event [24]. In addition, while older age has been found, in certain cases, to be more closely correlated with feelings of depression following exposure to a stressful event [7], age has also been found to be associated with the coping resources that individuals employ when they encounter a stressful event [25]. Many studies have found that teenagers use SOC as a coping resource following exposure to stressful events, and that this use of SOC varies by age [20]. Studies have found that there is a significant, if weak, relationship between age and sense of community resilience, such that as age increases, so does the sense of community resilience [26].

#### 1.4.2. Gender

Many studies have demonstrated the importance of gender as a factor that influences the intensity of reactions upon exposure to stress. Those studies have shown that, in general, women and girls exhibit greater injury following stressful events and are more likely to report stress symptoms and difficulties associated with internalization after such events. In contrast, men and boys tend to report more externalizing behaviors [20]. One possible explanation for this difference between genders is that women and girls react to stressful situations with more intense emotions than men and boys do, because they are more sensitive to the situation at hand [7,17]. In addition, it has been found that, in attempts to cope with a stressful situation, women and girls have a greater tendency to find social support and to rely on coping strategies that are focused on emotion and emotional control, as compared to men and boys, who tend to try to take control of the situation [27].

In the research literature, we find support for the idea that SOC is a protective factor during times of crisis, with this protective effect being stronger among men than it is among women [20]. When community resilience was examined as a coping resource, a relationship was observed between community resilience and gender, with women reporting higher levels of community resilience than men [26].

### 1.5. Community Resilience

The concept of community resilience includes the ability of a social system to hold, to react, and to tailor itself to crisis situations, to return to optimal functioning as soon as possible after a crisis, and to develop abilities that will allow it to cope with future crises [28]. The assumption that underlies the concept of community resilience is that people will prefer to work together to improve their community, as opposed to working as individuals [29]. Community resilience represents not only a community’s ability to cope with crises or disruptions that it may experience, but also the ability of the different systems within the community to preserve their relations with one another, their capabilities for cooperation, and their ability to recognize potential partners in the presence of the tensions and conflicts that arise during crises [30].

Communities that have high levels of community resilience exhibit a greater ability to cope with and overcome significant stressful situations and can recover more quickly from such situations, as compared to communities with lower levels of community resilience [22]. When examining the consequences of a person’s exposure to stressful situations, it is important to consider how that person perceives his/her community, the degree to which that person feels a sense of belonging to that community, and the level of the person’s exposure to the stress, as these factors may contribute to that person’s perceptions of community resilience and their subsequent reaction, coping, and adaptation in the context of the given stressful situation. The level of social cohesion among members of a community can contribute to the sense of community resilience [31]. Community resilience has been found to intensify residents’ sense of connection with the community in which they live, and develops the understanding that they can trust that the local authorities and their fellow residents will help them to cope with a crisis [28,32].

Surveys have shown that measures of community resilience among Bedouin society are much lower than those observed among general Israeli society [33] and among residents of Israeli communities located near the border with Gaza [34]. Large differences between communities have been observed in terms of trust in leadership and emergency preparedness [35]. Research conducted among the Arab population [36] that examined community resilience during a time of political violence (i.e., the Second Lebanon War) found powerful emotional reactions that stemmed from the local authorities’ and residents’ lack of preparedness for coping with a crisis situation.

### 1.6. Reactions to Stress among Teenagers Exposed to Stressful Events

The findings in the literature regarding the emotional reactions of teenagers to stressful events are not unequivocal. Most of the studies conducted among teenagers exposed to stressful situations have found different emotional symptoms: psychosomatic symptoms, post-traumatic reactions, etc. [37,38].

The research shows that, over the short and long term, individuals may experience emotional symptoms, generally expressed as depression and anxiety [39]; cognitive symptoms, expressed as confused thinking and difficulty thinking about the future [40]; and the development of serious stress-related diseases [41]. This has significant negative effects on teenagers’ daily routines and quality of life [9]. However, other studies have shown that resilience, resistance, and exposure to stressful events can lead to positive reactions, the development of social relationships, and improved self-perception [2,42]. The studies that point to teenagers’ resilience and resistance in situations of extreme stress support Antonovsky’s [43,44] approach. In conclusion, the literature points to a variety of emotional and cognitive reactions of teenagers to stressful situations, with some studies emphasizing the difficulties and negative symptoms, while others point to the potential for resilience and growth from the depths of a crisis [45].

### 1.7. Emotional Reactions of Bedouin Adolescents to Political Violence

The findings presented in the research literature discussing emotional reactions of Bedouin adolescents in stressful situations are not conclusive. Most studies conducted among Bedouin adolescents exposed to situations of political stress have found high levels of emotional reactions such as anger, anxiety, and psychological distress [7,46].

A study conducted in Israel during the Second Lebanon War, which compared stress reactions among Arab youth from northern Israel with those of Jewish youth living in the same region, found that adolescents from these different sectors reported similar, relatively low levels of stress [15]. One explanation for this is that in an acute-stress situation of war, differences between groups become blurred.

A study conducted in Israel following Operation Cast Lead (which occurred in 2009) showed that Bedouin adolescents reported more anger than Jewish adolescents [47]. A study examining emotional reactions and coping strategies among Bedouin adolescents following three military operations (Cast Lead in 2009, Pillar of Cloud in 2012, and Protective Edge in 2014) showed that Bedouin adolescents reported intense reactions of anger, anxiety, and psychological distress because, on the one hand, some of these adolescents had relatives and family members in Gaza (as a result of marriages with women from Gaza), while, on the other hand, they themselves were living under the enormous threat of hundreds of missiles, which were falling on the region, including their city, Rahat, in addition to the complex socioeconomic situation and other social challenges characteristic of Bedouin society [10].

### 1.8. Study Goals and Research Questions

This study aimed to examine how coping resources (i.e., personal SOC, community resilience, and the use of different coping strategies) explain differences in emotional reactions to stress (i.e., state anxiety, state anger, and psychological distress) among Bedouin teenagers, in the context of the events of October 7th and the Iron Swords War. This study was designed to examine the differences between boys and girls, in terms of the study variables, as well as the contributions of the different study variables to the explanation of the teenagers’ emotional reactions.

Specifically, this work examined the following research questions:Are there differences between boys and girls in terms of the study variables, namely, subjective exposure, community resilience, SOC, use of problem-solving and nonproductive coping strategies, state anxiety, state anger, and/or psychological distress?Are there relationships between the above-mentioned variables?Do any of the study variables explain the observed mental-health symptoms (i.e., anxiety, anger, and psychological distress)?

## 2. Materials and Methods

### 2.1. Data-Collection Procedure

Data were gathered between January and March 2024. Seven hundred and eighty questionnaires were physically distributed to potential participants, and 762 of those questionnaires were completed. The self-reported questionnaires asked about demographics, subjective exposure, community resilience, SOC, use of problem-solving, use of nonproductive coping strategies, state anxiety and anger, and psychological distress. No inclusion or exclusion criteria were applied other than age and availability at the time of testing.

All of the ethical procedures applicable to this study were followed. As required by the Israeli Ministry of Education, the Offices of the Central Scientist reviewed the research proposal and questionnaires before the start of the study (permit no. 13598). After receiving the approval to proceed with the study, we received permission from the principals to enter the schools. The students were informed that the researchers were interested in their experience of the events of October 7th and the ongoing war, that participation was voluntary and anonymous, and that they were free to withdraw their participation for any reason and at any time during the questionnaire procedure. The participants gave their consent a few days before they completed the questionnaires and they were also asked to give the consent forms to their parents. It took the participants 25–45 min to complete the questionnaire.

### 2.2. Participants

Data were collected from 762 participants, aged 12–18 (*M* = 16.40, *SD* = 0.93). Five hundred and twenty-one of the participants were girls (68.4% of the sample).

### 2.3. Measures

#### 2.3.1. Demographics and Exposure

Participants were asked to report their gender, age, and whether their mother and father were working. Participants were also asked about the number of missiles that had fallen in their community and the number of sirens that they had heard during the war.

#### 2.3.2. Subjective Exposure

Subjective exposure was assessed using five questions scored on a 4-point Likert scale. Adolescents reported their fear and sense of danger for themselves, their close and extended family, their friends, and the people in their community. Answers ranged from “not dangerous at all” to “very dangerous”. The sum of the scores for the answers to the questions was used as the subjective-exposure index. In the present study, the reliability of this measure was excellent; α = 0.90.

#### 2.3.3. Conjoint Community Resilience Assessment Measure (CCRAM; [48])

We used the short version of this scale, which includes 10 items that are each scored on a 5-point Likert-type scale ranging from 1 (do not agree at all) to 5 (definitely agree). The scale is constructed to assess CCRAM and to facilitate the estimation of an overall community-resilience score. It also detects the strength of five important constructs of community functioning following a disaster: leadership, collective efficacy, preparedness, place attachment, and social trust. Examples of items are: “I feel that I belong to the place where I live”; “I believe that my community has the ability to overcome a crisis”. The Cronbach’s alpha coefficient for the entire scale was α = 0.87.

#### 2.3.4. Sense of Coherence (SOC; [43])

SOC was measured using a series of semantic differential items rated on a 7-point Likert-type scale, with anchoring phrases at each end. High scores indicate a strong SOC. An account of the development of the SOC scale and its psychometric properties, showing it to be reliable and reasonably valid, appears in Antonovsky’s writings [43,49]. In this study, SOC was measured using the short-form scale consisting of 13 items, which has been found to be highly correlated to the original long version [49]. The scale includes such items as: “Doing the things you do every day is” with answers ranging from 1 (a source of pain and boredom) to 7 (a source of deep pleasure and satisfaction). In the present study, the Cronbach’s alpha coefficient was 0.62.

#### 2.3.5. Coping Strategies

Coping strategies were measured using the Adolescent Coping Scale (ACS; [50]). This instrument has been used to measure how adolescents cope with the experience of an actual or threatened home demolition. The short form comprises 18 items, which are each rated on a 5-point scale. The items are collapsed into three global coping styles: problem-solving, reference to others, and nonproductive coping. Since the “reference to others” scale is consistently unreliable in the Israeli context, the three scales were reduced to two scales, whose reliability was found to be very good: problem-solving (α = 0.78) and nonproductive emotional coping (α = 0.68).

#### 2.3.6. State Anxiety

The state anxiety measure, developed by Spielberger, Gorsuch, and Lushen [51] (Hebrew translation: [52]), was used to assess adolescents’ anxiety. The Hebrew translation has proven to be reliable, valid, and equivalent to the English State Anxiety Inventory [52]. This scale consists of 11 items, which are each rated on a 4-point Likert scale. Examples of questions are: “I feel peaceful”; “I am afraid of disasters”; and “I am worried”. The mean score was used, and the Cronbach’s alpha reliability coefficient was 0.77.

#### 2.3.7. State Anger

State anger was measured using the instrument developed by Spielberger et al. [51]. This scale consists of six items, which are each rated on a 4-point Likert scale (1—almost never, 4—almost always). Examples of questions include: “I am angry”; “I want to scream at someone”; and “I feel frustrated”. The mean score was used, and the Cronbach’s alpha reliability coefficient was 0.85.

#### 2.3.8. Psychological Distress

Psychological distress was evaluated using a five-item psychosomatic symptom scale, in which each item was rated on a 4-point Likert scale (1—never, 4—very frequently) referring to the frequency of occurrence of familiar symptoms (e.g., headaches, stomachaches). The scale was developed by Ben-Sira [53] and was later adapted for use among children [54]. In the present study, the Cronbach’s alpha coefficient was 0.81.

### 2.4. Data Analysis

Analyses were performed using SPSS version 28. We used *t*-tests for independent samples to check whether there were any significant differences between the boys and girls. We used Pearson correlations to examine the relationships between the different study variables. In addition, a hierarchical regression was run with demographic factors, gender, and age entered in the first step. The second step included the number of missiles that fell in the community, the number of sirens heard, and whether there was a bomb shelter in the participant’s home or neighborhood. Subjective exposure was entered in the third phase, community resilience was entered in the fourth step, and SOC was entered in the fifth step. Finally, in the sixth step, the coping strategies of problem-solving and nonproductive coping were entered.

## 3. Results

### 3.1. Findings Regarding Demographics and Exposure

Among the participants, 77% reported that their fathers were currently working, while 4.2% reported that their fathers were studying. The rest of the participants reported that their fathers were neither working nor studying. As for the mothers, 42.5% were working, 2.9% were studying, and the rest were neither working nor studying.

As for exposure to missiles and sirens that warn of attacks, 96% of the participants reported that they had been exposed to sirens in their neighborhood, and 79% reported that missiles had fallen in their community. Fifty-two percent of the participants reported that they had a bomb shelter in their neighborhood.

### 3.2. Research Question 1: Were There Differences between Boys and Girls, in Terms of the Study Variables?

To answer the first question, a *t*-test for independent samples was run. The results are presented in Table 1. These results show significant differences in several variables, namely, subjective exposure, use of problem-solving strategies, use of nonproductive coping strategies, state anxiety, state anger, psychological distress, and mental-health. In all cases, girls reported higher scores than boys. This indicates that the girls felt more subjective exposure and danger, used a greater variety of coping strategies, and reported higher levels of different forms of psychological distress.

### 3.3. Research Question 2: Were There Relationships between the Study Variables?

To answer the second question, Pearson correlations were run. Most of the examined correlations were found to be significant. Medium-strength relationships were found between the use of nonproductive coping strategies and negative mental-health outcomes (i.e., state anger, state anxiety, and psychological distress), between SOC and negative mental-health outcomes, as well as between subjective exposure and negative mental-health outcomes, as shown in Table 2.

### 3.4. Research Question 3: Which of the Study Variables Were Associated with Mental-Health Outcomes?

The third research question related to the explanation of mental-health outcomes by the different demographic variables, subjective exposure, and personal and community resources, as well as the use of different coping strategies. Mental-health scores were calculated by computing the mean of three variables: anxiety, anger, and psychological distress. Hierarchical multiple regression was performed in six steps. The findings regarding the relationships between each variable and the examined mental-health problems are presented in Table 3. In the first step of the analysis, gender and age explained 6% of the variance, and both were significant in their explanation. The second step added 3% to the explained variance, with number of missiles falling in the neighborhood and having a shelter in the neighborhood as significant explanatory factors. In the third step, subjective exposure added 7% to the explanation of the variance; community resilience added 1% in the fourth step. SOC added 15%, and problem-solving and nonproductive coping added an additional 11% in the sixth step. All were significant in their explanation. Thus, the entire model explained 43% of the variance in the examined mental-health problems.

## 4. Discussion

The goal of this study was to examine the levels of subjective exposure, coping strategies, and emotional reactions to stress (i.e., anxiety, anger, and psychological distress) among Bedouin Arab teenagers following the events of October 7th and during the Iron Swords War. We found that, in general, all of the Bedouin teenagers reported strong emotional reactions to the stress. The war situation, characterized by ambiguity and uncertainty; the suddenness and unexpectedness of the events; and the ongoing danger from missile-fire contributed to psychological distress and high levels of anger. With this finding, this work joins a series of studies conducted among teenagers exposed to stressful events that have found high levels of emotional symptoms, both around the world [45,55] and in Bedouin society [7,46].

Another explanation for the high levels of emotional reactions relates to the realities of life in Bedouin society, which is characterized by a dire lack of physical, economic, social, and communal resources [42]. The greater vulnerability of Bedouin society is related to its low socioeconomic status, the lack of access to health services, cultural barriers, communal isolation, and a lack of trust in governmental institutions [56]. In addition, the fact that the enemy speaks the same language and shares a similar history may intensify the emotional reactions to the stressful situation [34].

We found that the girls reported more intense subjective exposure than the boys did. There are several explanations for this finding. It may be related to internal dialogue regarding the level of threat inherent in the situation, which has been found to be more intense among girls [57]. Another explanation is that the girls experienced a greater sense of helplessness, apparently due to their limited involvement in finding solutions and helping others. This may have limited their sense of control over the situation [58]. Another explanation refers to the culture of Bedouin society, in which social and cultural prohibitions can limit girls’ connections with professionals. As a result, they are less exposed to information about the management of the crisis, which may intensify their fears and concerns [59].

The boys and girls also differed in their use of coping strategies. The girls made greater use of both problem-solving strategies and nonproductive strategies. That finding corresponds with those of previous studies [60,61]. The girls may have made greater use of problem-solving strategies because they tend to receive more social support [25]. The sociological-cultural assumption that girls talk and share more with each other may be a self-fulfilling prophecy, which allows the girls to make greater use of problem-solving strategies. A study conducted among Arab society found that girls are more likely to turn to mental-welfare resources at school and to participate in social-enrichment programs [59]. Those resources and programs may encourage the use of problem-solving strategies [62]. In contrast, a sense of lack of control among the boys may reduce their use of problem-solving strategies. We can suggest a number of possible explanations for the girls’ greater use of emotion-focused strategies. The first explanation concerns social expectations. The society may expect girls to be more “emotional” and “empathetic”, which may encourage their use of emotion-focused strategies [63]. An additional cultural explanation is that emotional strategies are generally perceived as “gentle” and “soft”, qualities that societies like Bedouin society do not associate with traditional masculinity. As a result, boys may avoid using these strategies, based on a fear of being perceived as insufficiently manly [64].

The girls also reported more state anger, state anxiety, and psychological distress. This finding is congruent with the findings of previous studies that have examined gender differences in reactions to stress in the context of exposure to stressful events [19,65,66]. The common explanation for these gender differences is that girls react to stressful events with more intense emotion, since they are more sensitive to situations. An additional cultural explanation for the observed gender differences is that girls have greater social permission to report weakness and difficulty than boys do. Therefore, they may tend to report their reactions to stress more explicitly. However, receiving help and support from the community is associated with neediness, which is less acceptable in Arab society [42]. Limitations on the mobility of women in Bedouin society may intensify their emotional reactions, as they intensify the feelings of psychological distress due to actual danger.

The second research question referred to the relationship between the independent study variables and the dependent study variables. We hypothesized that the independent variables (i.e., subjective exposure, community resilience, personal SOC, and use of effective and ineffective coping strategies) would be associated with the emotional reactions to stress. Subjective exposure was associated with emotional reactions in a significant manner, as reported in previous studies [67]. It is reasonable and makes sense that physical proximity to the site of a disaster and the teenager’s acquaintance with severely affected others (i.e., the fallen, hostages, and their families) would give rise to emotional reactions [19]. Among the Bedouin teenagers, the lack of protective equipment (i.e., sirens, communal bomb shelters, and portable bomb shelters) and support resources (i.e., lack of welfare resources in the communities) significantly amplified the sense of anger, the sense of anxiety, and the psychological distress.

It is important to note that the Arab society in Israel suffers from a high crime rate and, specifically, a relatively high murder rate, as well as a dire socioeconomic situation in which 60% of households live below the poverty line [5]. The ongoing exposure to political violence, missile-fire, and the threat of continued war, in addition to the stress of normal times, have intensified the feelings of psychological distress, state anger, and state anxiety among these teenagers. Another explanation is the ambiguity, uncertainty, and helplessness that characterize the situation, in which citizens do not know when they can expect the war and the threat to come to an end [68]. Another reason is the threat to the future, the future in the state of Israel, and the relationship with Israeli society [7].

Regarding the research question about the relationships among the different variables, this study found an inverse relationship between personal SOC and emotional reactions: the stronger the personal SOC, the lower the levels of anxiety, anger, and psychological distress. A number of explanations can be offered for this finding. The first is that of cultural influence. The culture of Bedouin society may demand that teenagers demonstrate resilience and resistance, as an expression of their loyalty to their tribe and their society. This could lead to the reporting of high levels of personal SOC. Another explanation is related to the support that the teenagers received from their schools and from social services [69].

A third explanation for the association between personal sense of coherence and the examined emotional reactions relates to the role of religious faith, which is a central component of Bedouin society and could contribute significantly to SOC. Religion can contribute to each of the components of SOC. In terms of comprehensibility, religion provides an explanatory framework for life events, including stressful situations and crises, and offers explanations and meanings for difficult events, which can help individuals to make sense of such events. In terms of manageability, the belief in a higher power and spiritual support can strengthen the sense that one is capable of dealing with challenges. Religious practices, such as prayer or meditation, can serve as coping tools. Finally, in terms of meaningfulness, religion provides a sense of a goal and the significance of life, even during difficult times. It can help people to see events as part of a larger picture [9].

We also examined how the different study variables were associated with the observed emotional reactions. We found that younger teenagers reported more intense emotional reactions. This finding was previously reported in other studies [44,70]. One of the possible explanations for this finding relates to cognitive development. We can assume that the cognitive development of the older teenagers has expanded their ability to understand the severity of the situation. Therefore, they feel that they possess the means to cope with the situation and exhibit fewer feelings of distress than younger teenagers [25]. Due to their age and level of emotional maturity, younger children express greater distress [71]. In addition, in terms of societal expectations, older teenagers are expected to hide their feelings [18].

Subjective exposure also contributed to the explanation of the emotional reactions, and this makes sense because physical proximity to the site of a disaster and acquaintance with some of the victims (i.e., hostages, the fallen, and their families) would be expected to give rise to strong emotional reactions [72]. The subjective-exposure variable also made a large contribution to the intensity of the emotional reactions. One explanation for this is that the fact that Bedouin society is intensely exposed to political crises (e.g., home demolitions), and social crises (e.g., social violence, crime, complex socioeconomic situation, lack of basic protective equipment) may increase the intensity of the emotional reactions and the fear of significant harm to property and human life [73].

The findings regarding coping strategies provide support for the findings of previous studies [15,19], which reported that both problem-solving strategies and nonproductive strategies are significant for the explanation of emotional reactions. Whereas the use of problem-solving strategies decreased the intensity of the emotional reactions; the use of nonproductive strategies was associated with more intense emotional reactions. This may be related to the fact that the teenagers, following their use of emotion-focused strategies, may have trouble constructing a more positive conceptual alternative explanation for a traumatic event, in addition to the belief that the event was not controllable [7]. Problem-solving strategies may be congruent with the values of Bedouin society (e.g., sense of control and resilience) and Islam, which is the dominant religion in this society, in contrast to nonproductive coping strategies that may broadcast weakness, which is not socially acceptable [9]. An additional explanation could be that the teenagers turned to authority figures within their families (e.g., parents) or schools (e.g., school counselors), and that action decreased their feelings of stress.

Finally, personal SOC was associated with the emotional reactions in the most significant way. We found that the higher the level of personal SOC, the less intense the emotional reactions. This finding underscores the importance of SOC as a protective psychological factor. Although this has been found frequently in Western populations, it is surprising in the context of Bedouin society and among Bedouin teenagers, in particular. Previous studies of SOC among Bedouin Arabs did not find that it made any significant contribution to the explanation of psychological distress, and some studies even found that SOC is not a protective factor for the Bedouin population [7,15,74]. One explanation for this is the social changes that have occurred since 2015 (when the first of the studies cited above was conducted). In recent years, there have been many political and cultural changes in Bedouin society, in addition to global events (e.g., the COVID-19 pandemic). These changes and others have caused the society to move from a patriarchal-collectivist model to a more individualistic-Western model. Other possible explanations are related to increased awareness, access to information sources and social media, and the development of emotional discourse among Bedouin teenagers [75]. Another possible explanation is that with changes in the war situation (i.e., the ground invasion of Gaza and the decrease in missile-fire), the teenagers have accumulated experience and begun to become accustomed to the situation. The installation of portable bomb shelters in the unrecognized villages, the decreased missile-fire, and the return to school and meetings with social-service professionals in the schools may have given the teenagers a greater sense of stability [42,75].

### 4.1. Study Limitations

This research relied on self-reports, which may be biased. In addition, data were collected at one point in time, without any long-term monitoring that could have provided a more comprehensive picture of long-term influences. With that, the advantage of this study is that it was conducted in real time, under the pressure of war. Additionally, we did not examine additional factors that might influence emotional reactions, such as familial support or specific socioeconomic status. The final limitation is the lack of a control group of teenagers who were not exposed to the war events, which could sharpen our understanding of the influence of those events. These limitations should be addressed in future studies.

### 4.2. Conclusions and Recommendations

This study examined the emotional reactions and coping resources of Bedouin teenagers (ages 12–18) following the events of October 7th and the Iron Swords War. The findings show that the girls reported higher levels of subjective exposure and more intense emotional reactions than the boys. The girls also made greater use of both coping strategies based on problem-solving and nonproductive coping strategies. Subjective exposure was significantly associated with the observed emotional reactions. Personal SOC was found to be a protective factor, whereas the use of nonproductive coping strategies intensified psychological distress.

Based on these findings, we make the following recommendations for the educational system and schools that serve Bedouin Arab students:Develop gender-tailored intervention programs: programs that take into consideration gender differences in emotional reactions and coping strategies should be developed for boys and girls.Train educational staff: educational professionals should be trained in the early detection of emotional reactions to stress and taught how to offer coping tools, including ways to enhance personal SOC, with consideration given to the unique cultural characteristics of Bedouin society, including recent and ongoing changes in that society.

This research highlights the importance of understanding gender differences and the factors that influence emotional reactions to stressful situations among Bedouin teenagers. The application of our recommendations could help to improve how Bedouin Arab teenagers cope with stressful situations and promote their mental welfare.

## Figures and Tables

**Table 1 behavsci-14-00900-t001:** Gender differences in the study variables.

Variables	All *N* ≈ 756		Boys *N* ≈ 238		Girls *N* ≈ 516		*t*
	M	SD	M	SD	M	SD	
Subjective exposure (range 6–36)	21.07	6.40	19.48	6.64	21.83	6.15	−4.75 ***
CCRAM (range 1–5)	2.80	0.94	2.79	1.03	2.81	0.90	−0.22
SOC (range 1–7)	4.06	0.83	4.12	0.80	4.02	0.83	1.56
Problem-solving (range 18–90)	56.23	17.08	54.29	18.42	57.11	16.39	−2.11 *
Nonproductive coping (range 18–90)	49.51	13.33	46.05	14.20	51.08	12.61	−4.88 ***
State anxiety (range 1–4)	2.40	0.60	2.23	0.57	2.47	0.59	−5.16 ***
State anger (range 1–4)	2.07	0.82	1.94	0.84	2.12	0.80	−2.94 **
Psychological distress (range 1–4)	2.33	0.84	2.13	0.91	2.33	0.79	−4.16 ***
Mental health (range 1–4)	2.25	0.60	2.09	0.61	2.33	0.58	−5.06 ***

*** *p* < 0.001; ** *p* < 0.01; * *p* < 0.05.

**Table 2 behavsci-14-00900-t002:** Correlations between the study variables.

	1	2	3	4	5	6	7	8
1. Subjective exposure	-	0.18 ***	−0.16 ***	0.17 ***	0.31 ***	0.29 ***	0.22 ***	0.26 ***
2. CCRAM		-	0.16 ***	0.28 ***	0.12 ***	0.11 ***	−0.06	0.01
3. SOC			-	0.14 ***	−0.13 ***	−0.37 ***	−0.36 ***	−0.36 ***
4. Problem-solving				-	0.54 ***	−0.10 ***	0.03	−0.01
5. Nonproductive coping					-	0.31 ***	0.37 ***	0.25 ***
6. State anxiety						-	0.67 ***	0.37 ***
7. State anger							-	0.33 ***
8. Psychological distress								-

*** *p* < 0.001.

**Table 3 behavsci-14-00900-t003:** Results of hierarchical multiple regression analysis predicting mental-health problems.

	Mental-Health Problems				
	*R^2^*	*B*	*β*	*SE*	*t*
Step 1	0.06				
Gender		0.27	0.21	0.05	5.57 ***
Age		0.09	0.14	0.02	3.66 ***
Step 2	0.03				
Gender		0.28	0.22	0.05	5.94 ***
Age		0.08	0.11	0.03	2.94 **
Sirens		0.01	0.07	0.01	1.88
Missiles		0.01	0.08	0.01	2.06 *
Shelter		−0.14	−0.11	0.04	−3.03 **
Step 3	0.07				
Gender		0.21	0.17	0.05	4.50 ***
Age		0.05	0.08	0.02	2.15 *
Sirens		0.01	0.05	0.01	1.33
Missiles		0.01	0.06	0.01	1.42
Shelter		−0.11	−0.10	0.04	−2.66 **
Subjective exposure		0.03	0.00	0.28	7.46 ***
Step 4	0.01				
Gender		0.20	0.16	0.05	4.34 ***
Age		0.05	0.07	0.02	1.97 *
Sirens		0.01	0.04	0.01	1.54
Missiles		0.01	0.06	0.01	1.47
Shelter		−0.09	−0.07	0.04	−2.04 *
Subjective exposure		0.03	0.30	0.00	8.00 ***
CCRAM		−0.07	−0.11	0.02	−3.05 **
Step 5	0.15				
Gender		0.18	0.14	0.04	4.13 ***
Age		0.03	0.05	0.02	1.51
Sirens		0.01	0.05	0.00	1.34
Missiles		0.01	0.03	0.01	0.79
Shelter		−0.10	−0.08	0.04	−2.38 *
Subjective exposure		0.02	0.24	0.00	6.84 ***
CCRAM		−0.02	−0.03	0.02	−0.97
SOC		−0.29	−0.40	0.02	−11.91 ***
Step 6	0.11				
Gender		0.11	0.09	0.04	2.84 **
Age		0.04	0.06	0.02	1.83
Sirens		0.00	0.03	0.00	1.05
Missiles		0.00	0.02	0.01	0.77
Shelter		−0.07	−0.06	0.04	−1.91
Subjective exposure		0.02	0.18	0.00	5.32 ***
CCRAM		−0.01	−0.02	0.02	0.45
SOC		−0.23	−0.32	0.02	−9.99 ***
Problem-solving		−0.01	−0.24	0.00	−6.41 ***
Nonproductive coping		0.02	0.42	0.00	11.14 ***

*** *p* < 0.001; ** *p* < 0.01; * *p* < 0.05.

## Data Availability

Data are available upon request.

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
