# Peer review of "Bedouin Adolescents during the Iron Swords War: What Strategies Help Them to Cope Successfully with the Stressful Situation?"

_behavsci, 2024, doi:10.3390/bs14100900_

Round 1

Reviewer 1 Report

Comments and Suggestions for Authors

The paper presents a study that examined the emotional reactions and coping resources of Bedouin teenagers in southern Israel following the events of October 7th, 2023 and the Iron Swords War. The study explored subjective exposure, personal sense of coherence, community resilience, and the use of effective and non-effective coping strategies. It also examined differences between boys and girls in terms of the study variables.

This topic is important because it deals with a vulnerable population in a very complicated situation. The Bedouin community in Israel suffers from lower socioeconomic status, communal isolation, distrust of government institutions, and cultural barriers such as the stigma against boys and men seeking mental health support. Previous studies conducted among Bedouin teenagers showed that when exposed to stressful events, they reacted with high levels of emotional symptoms. The current situation is especially complicated for them because they were victims of attacks – missiles, abduction, and close-range gunfire – by people who share a similar historical, linguistic, and cultural background as them.

The literature review appropriately analyzes previous research, most of it recent. The literature is used well to generate hypotheses.

The methods section clearly describes the sample, the variables included in the questionnaire, and the statistical tests used in the analysis. In terms of findings, unlike in previous studies, personal sense of coherence was found to significantly increase the ability to cope with stress.

The conclusions include recommendations to develop intervention programs that take into consideration gender differences and the factors that influence emotional reactions to stressful situations among Bedouin teenagers. In addition, they recommend training educational staff members to help them to identify and treat emotional reactions to stress in culturally-appropriate ways. These conclusions are well-supported by the results.

The article is well written. However, the complexity of the situation for Bedouin teenagers, while touched upon, should be emphasized even more.

Author Response

Dear Editor,

We would like to express our sincere gratitude to both reviewers for their thoughtful and insightful comments on our manuscript. Their feedback has significantly sharpened and enriched our paper. We appreciate the time and effort they have invested in reviewing our work.

Below, we provide a point-by-point response to each of the reviewers' comments and concerns. We have carefully considered all suggestions and have made appropriate revisions to our manuscript accordingly.

Point-by-point responses:

Reviewer 1

Reviewer Comment

Response and Revision

Page Number

the complexity of the situation for Bedouin teenagers, while touched upon, should be emphasized even more.

Please see our revised Section 1.2.

Pages 2-3

Lines 62-109

Reviewer 2 Report

Comments and Suggestions for Authors

Thanks for the opportunity to review the article Bedouin Adolescents During the Iron Swords War:  Which Coping Mechanisms Help Them To Cope Successfully With the Stressful Situation? which addresses a topic that is associated with a problematic situation that needs to be addressed, especially about teenagers and their well-being. The article provides stimulating and essential information and is very well done, presenting the data clearly and providing exciting interpretations of the findings.

In the comments below, I reflect on a few points that could have been more obvious and would therefore recommend clarifying them.

Abstract - it talks about the effect "the study focused on effects of ..." Given the study's design, which was not longitudinal, the word effect may be problematic as it demonstrated a correlation and relationship between variables.

Introduction

The introduction to the topic is adequate, in places quite detailed (especially in the description of community resilience, etc.).

Small additions are appropriate to help better understand the situation, which is evident to the writers, but readers may not be familiar with the outline:

Line 38 - the government evacuated = Israel government?

line 47 - from well-known families in the community = in the Bedouin or Israel community

Conceptually unify (put in a common subchapter?) the information in lines 42-69 - all this information is related to Bedouin Arab society. Still, some critical explanatory information is only in subchapter 1.2 (which, however, is not only about teenagers but about the Bedouin community in general, so this subheading is a bit confusing).

- It would be helpful to give information about the Bedouins in general and how they are affected by the current war. But I understand that it depends on the overall concept of the authors

Missing subchapter 1.3 - is it a case of inattention?

- that could appropriately bridge information about the war, the community, and just information about what research already knows about Bedouin teenagers' reactions during periods of crisis and why it is good to research further

Line 83 - it would be helpful to provide a reference supporting this claim

Line 91 - in what contexts are references to any research that supports this?

Line 94 - this sentence would be helpful to elaborate slightly; although a reference is given, the sentence cannot be generalized on that basis. Where does the Arab community suffer? In Israel or globally? Given the context of the war and the terrorist attack, the Jewish community is also suffering... I recommend considering refining the sentence, and similarly, if in the following lines 94-96, the authors are talking about studies (plural), it is appropriate to give at least two references.

In subsection 1.5, some information on community resilience is redundantly repeated. It would be appropriate to shorten the subchapter slightly and include only the information relevant to the study - especially those at the end of the subchapter, similar to Sociodemographic factors. In this context, subsection 1.7 seems like the one that should follow up subsection 1.4 on coping, which ends with just the sentence on emotional reactions.

The subchapters jump around a bit in terms of ideas and do not follow - the authors summarise this nicely in lines 208-210, where they show what is related to what.

Lines 213 and 216 should be phrased as the goal/purpose of the study, not as a done deal, past tense "the study examined"

Materials and Methods

Data collection - how was the data collected? Online? Was there an inclusion/exclusion criteria for the study?

Did the final count of 762 respondents include all those who completed the questionnaire, or were there some respondents who had to be excluded (incomplete data, incomplete, etc.)?

Since I am interested in young individuals, were the parents/legal guardians of the children informed about the study and its purpose? It is mentioned in the informed consent, but at least a mention that the parents were informed would be appreciated here.

The information on lines 245-254 belongs more in the results.

Line 257 - the number of missiles and sirens is not a demographic measure

For subjective exposure, was there a median response? If not, why not?

Line 267 - In the present study, α = .90. It seems like an incomplete sentence, similarly 299

Data analysis - In what program was the analysis conducted? What analyses were performed in steps two through six? The authors only state which variable was entered, but it is unclear which model was entered. Yet, in the descriptions of the results, information is sometimes given that falls more into this part of the

Results - The results are well-described

Discussion

This section is well done, reads well, and interprets the study's findings appropriately. I appreciate the inclusion of sociocultural factors.

There are only a few points that would be good to clarify

Line 470 - it would be helpful to give a reference to that study

In the paragraph from line 498 onwards, I would be cautious about using the word "explain." the study showed a relationship between variables, but based on the study's design, it is difficult to talk about causality as much as correlation association, etc. Similarly line 543

It would be helpful to add a reference to the parenthetical remark on line 514 to a statement that seems quite fundamental and unambiguous

If the sentence on line 531 begins as a separate paragraph, it would be appropriate to repeat the explanation of what. In the religion paragraph, I recommend adding a reference to prove the rather fundamental claim.

The information about the transformation of the Bedouin Society on lines 595-599 is very interesting. In terms of the recommendations in the conclusion of the study, should this also not be somehow reflected in the education system and work with the Bedouin community, as these may be factors that will continue to be strongly associated with the SOC of Bedouin teenagers and the community as a whole in the future.

After reading the entire article and especially the results, I also went back to the title of the article Bedouin Adolescents During the Iron Swords War: Which Coping Mechanisms Help Them To Cope Successfully With the Stressful Situation? It seems the results do not so much talk about coping mechanisms per se but somewhat different strategies (SOC, anger....) that help with coping. Didn't the authors consider titling the paper more obsessively to reflect the outcomes? For example, Bedouin Adolescents During the Iron Swords War: What (what strategies) Help Them To Cope Successfully With the Stressful Situation?

Congratulations to the authors for a study on an interesting topic that, while focusing on a small group of teenagers in a specific war situation, also provides essential insights into the coping strategies of youth in general and which factors to pay attention to in the event of a threat (which hopefully will end soon).

Comments on the Quality of English Language

Sometimes, there are sentences that are more difficult to understand because of their style. In some places, the pronouns "this" and "these" and what exactly they refer to are also difficult to understand.

Author Response

Reviewer 2

Reviewer Comment

Response and Revision

Page Number

Abstract

It talks about the effect "the study focused on effects of ..." Given the study's design, which was not longitudinal, the word effect may be problematic as it demonstrated a correlation and relationship between variables.

We now refer to relationships instead of effects.

Page 1

Line 14

Introduction

Line 38 - the government evacuated = Israel government?

Yes, the Israeli government.

Page 1

Line 38

Line 47 - from well-known families in the community = in the Bedouin or Israel community

The text has been changed to read “well-known families in the Bedouin community.”

Page 2

Line 47

Conceptually unify (put in a common subchapter?) the information in lines 42-69 - all this information is related to Bedouin Arab society. Still, some critical explanatory information is only in subchapter 1.2 (which, however, is not only about teenagers but about the Bedouin community in general, so this subheading is a bit confusing).

Please see our revised Section 1.2.

Pages 2-3

Lines 62-109

Missing subchapter 1.3 - is it a case of inattention? that could appropriately bridge information about the war, the community, and just information about what research already knows about Bedouin teenagers' reactions during periods of crisis and why it is good to research further

In response to this comment, we added an additional subsection to the Introduction: “1.7. Emotional Reactions of Bedouin Adolescents to Political Violence.”

Pages 5-6

Lines 244-278

Line 83 - it would be helpful to provide a reference supporting this claim

We added a reference to support this claim.

Page 6

Line 258

Line 91 - in what contexts are references to any research that supports this?

We added a reference.

Page 3

Line 139

Line 94 - this sentence would be helpful to elaborate slightly; although a reference is given, the sentence cannot be generalized on that basis. Where does the Arab community suffer? In Israel or globally? Given the context of the war and the terrorist attack, the Jewish community is also suffering... I recommend considering refining the sentence, and similarly, if in the following lines 94-96, the authors are talking about studies (plural), it is appropriate to give at least two references.

Please see the revised Section 1.7 and the references in that section.

Pages 5-6

Lines 244-278

In subsection 1.5, some information on community resilience is redundantly repeated. It would be appropriate to shorten the subchapter slightly and include only the information relevant to the study - especially those at the end of the subchapter, similar to Sociodemographic factors. In this context, subsection 1.7 seems like the one that should follow up subsection 1.4 on coping, which ends with just the sentence on emotional reactions.

The subchapters jump around a bit in terms of ideas and do not follow - the authors summarise this nicely in lines 208-210, where they show what is related to what.

We revised the section on community resilience (Section 1.5) to minimize redundancy.

In response to this comment, we also re-arranged the subsections in the Introduction.

Pages 4-5

Lines 184-222

Pages 1-6

Lines 29-285

Lines 213 and 216 should be phrased as the goal/purpose of the study, not as a done deal, past tense "the study examined

We rephrased this sentence.

Page 6

Lines 244-275

Materials and Methods

Data collection - how was the data collected? Online? Was there an inclusion/exclusion criterion for the study?

The data were collected through research questionnaires that were physically distributed among the teenagers. No inclusion or exclusion criteria were applied other than age and availability at the time of testing. This information was added to the manuscript.

Page 6

Lines 289-295

Did the final count of 762 respondents include all those who completed the questionnaire, or were there some respondents who had to be excluded (incomplete data, incomplete, etc.)?

We distributed 780 questionnaires and 762 questionnaires were completed. This information was added to the manuscript.

Page 6

Lines 288-290

Since I am interested in young individuals, were the parents/legal guardians of the children informed about the study and its purpose? It is mentioned in the informed consent, but at least a mention that the parents were informed would be appreciated here.

Certainly, the teenagers gave their consent a few days before they filled out the questionnaires and were asked to give the consent forms to their parents. This information was added to the manuscript.

Page 7

·       Lines 305-307

The information on lines 245-254 belongs more in the results.

The findings from the demographic survey are now presented in Section 3.1.

Page 9

Lines 399-409

Line 257 - the number of missiles and sirens is not a demographic measure

This information is now presented in Section 3.1.

Page 9

Lines 405-409

Line 267 - In the present study, α = .90. It seems like an incomplete sentence, similarly 299

Corrected.

Page 7, Line 326

Page 8, Line 359

Data analysis - In what program was the analysis conducted? What analyses were performed in steps two through six? The authors only state which variable was entered, but it is unclear which model was entered. Yet, in the descriptions of the results, information is sometimes given that falls more into this part of the

Please see the revised Section 2.4.

Page 9

Lines 385-397

Discussion

Line 470 - it would be helpful to give a reference to that study

We added a reference.

Page 13

Line 532

In the paragraph from line 498 onwards, I would be cautious about using the word "explain." the study showed a relationship between variables, but based on the study's design, it is difficult to talk about causality as much as correlation association, etc. Similarly line 543

Throughout the Discussion, we now use the term associated instead of explained.

It would be helpful to add a reference to the parenthetical remark on line 514 to a statement that seems quite fundamental and unambiguous.

In response to this comment, we added a sentence with a reference.

Page 14

Lines 566-569

If the sentence on line 531 begins as a separate paragraph, it would be appropriate to repeat the explanation of what. In the religion paragraph, I recommend adding a reference to prove the rather fundamental claim.

We added the requested explanation and a reference.

Page 14

Lines 587-588, 600

The information about the transformation of the Bedouin Society on lines 595-599 is very interesting. In terms of the recommendations in the conclusion of the study, should this also not be somehow reflected in the education system and work with the Bedouin community, as these may be factors that will continue to be strongly associated with the SOC of Bedouin teenagers and the community as a whole in the future.

We revised our second recommendation accordingly.

Page 16

Lines 692-696

After reading the entire article and especially the results, I also went back to the title of the article Bedouin Adolescents During the Iron Swords War: Which Coping Mechanisms Help Them To Cope Successfully With the Stressful Situation? It seems the results do not so much talk about coping mechanisms per se but somewhat different strategies (SOC, anger....) that help with coping. Didn't the authors consider titling the paper more obsessively to reflect the outcomes? For example, Bedouin Adolescents During the Iron Swords War: What (what strategies) Help Them To Cope Successfully With the Stressful Situation?

The title of the manuscript was changed as suggested by the reviewer.

Page 1

Line 3

Thanks again for all the comments and insights that enriched the article. We hope the article will contribute to the research literature, and we all hope that the war will end soon and that everyone will live in peace.
